# A Novel Thiophene-Based Fluorescent Chemosensor for the Detection of Zn^2+^ and CN^−^: Imaging Applications in Live Cells and Zebrafish

**DOI:** 10.3390/s19245458

**Published:** 2019-12-11

**Authors:** Min Seon Kim, Dongju Yun, Ju Byeong Chae, Haeri So, Hyojin Lee, Ki-Tae Kim, Mingeun Kim, Mi Hee Lim, Cheal Kim

**Affiliations:** 1Department of Fine Chemistry, Seoul National University of Science and Technology, Seoul 01187, Korea; dltmf2303@naver.com (M.S.K.); juju9593@hanmail.net (D.Y.); ch920812@naver.com (J.B.C.); gofl0988@naver.com (H.S.); 2Department of Environmental Engineering, Seoul National University of Science and Technology, Seoul 01187, Korea.; hyojin_lee@seoultech.ac.kr; 3Department of Chemistry, Korea Advanced Institute of Science and Technology, Daejeon 34140, Korea; mingeun@unist.ac.kr (M.K.); miheelim@kaist.ac.kr (M.H.L.)

**Keywords:** fluorescence chemosensor, zinc, cyanide, living cell, zebrafish

## Abstract

A novel fluorescent turn-on chemosensor **DHADC** ((E)-3-((4-(diethylamino)-2-hydroxybenzylidene)amino)-2,3-dihydrothiophene-2-carboxamide) has been developed and used to detect Zn^2+^ and CN^−^. Compound **DHADC** displayed a notable fluorescence increase with Zn^2+^. The limit of detection (2.55 ± 0.05 μM) for zinc ion was far below the standard (76 μM) of the WHO (World Health Organization). In particular, compound **DHADC** could be applied to determine Zn^2+^ in real samples, and to image Zn^2+^ in both HeLa cells and zebrafish. Additionally, **DHADC** could detect CN^−^ through a fluorescence enhancement with little inhibition with the existence of other types of anions. The detection processes of compound **DHADC** for Zn^2+^ and CN^−^ were demonstrated with various analytical methods like Job plots, ^1^H NMR titrations, and ESI-Mass analyses.

## 1. Introduction

The design of chemosensors with high selectivity and sensitivity has received great interest because they can recognize environmentally and biologically crucial metal ions and anions [1,2]. Among these ions, zinc ion is not only one of the essential metal ions in the human body, but is also the second richest transition metal ion [3,4]. It has large roles at catalytic sites of myriad Zn^2+^-containing metalloenzymes and in DNA-binding proteins [5,6]. Meanwhile, an uncontrolled zinc concentration in the body creates a wide variety of troubles like epilepsy, Parkinson disease, and ischemic stroke [7]. Hence, it is of great significance to design chemosensors for the selective sensing of Zn^2+^ in biological systems [8,9,10,11].

Recently, anions have attracted notable interest, owing to their various roles in clinical, environmental, and biological applications [12,13,14]. In particular, cyanide has been extensively used in numerous territories like synthetic fiber, gold mining, resin industries, and metallurgy [15,16,17,18]. Thus, the voluminous usage of cyanide is ineludible, and numerous industries yield about 140 k tons/year of cyanide [19,20,21]. However, cyanide acts as a strong poison. Its toxicity induces the susceptibility of binding to iron ion in metalloprotein cytochrome oxidase, blocking the electron transfer chain in mitochondria [22,23,24]. Moreover, high levels of cyanide can cause convulsions, vomiting, loss of consciousness, and ultimately death [25,26]. Thus, it is essential to develop an effective sensing tool to recognize the cyanide level in living organisms and environments [27,28].

Among various analytical methods, a fluorescent method has attracted much attention due to its high selectivity, simplicity, and bioimaging ability [29,30,31,32]. Until now, a few fluorescence chemosensors for detecting both Zn^2+^ and CN^−^ were developed, but they are still rare. In addition, zinc fluorescent chemosensors for bioimaging in living cells and zebrafish are very rare (Appendix A) [33,34,35,36,37,38,39]. Therefore, the development of fluorescent chemosensors with high selectivity and bioimaging ability in both living cells and zebrafish is needed.

Thiophene derivatives have been extensively utilized as a fluorescence signaling promoter to anions, organic acids, and metal ions [40,41]. Moreover, 4-diethylaminosalicylaldehyde moiety is an outstanding fluorophore that has a water-soluble electron-donor property [42,43]. Thus, we combined the two functional groups to design a novel and practical fluorescent sensor, which is expected to sense a particular analyte through a unique fluorescent property with bioimaging ability in both living cells and zebrafish.

Here, we demonstrate a novel and stable fluorescent chemosensor **DHADC,** comprised of 3-aminothiophene-2-carboxamide as a fluorescence-signaling group and 4-diethylaminosalicylaldehyde as an electron-donating group (Scheme 1). Chemosensor **DHADC** detected both Zn^2+^ and CN^−^ by fluorescent turn-on. To interpret their detecting systems, diverse analytical investigations like ESI-mass analyses, ^1^H NMR titrations, and Job plots were carried out.

## 2. Experimental

### 2.1. Reagents and Equipments

Chemicals were purchased from Sigma–Aldrich. A Varian spectrometer was used to obtain ^13^C (100 MHz) and ^1^H NMR (400 MHz) spectra. Fluorescence emission and UV-visible absorption spectra were recorded with Perkin Elmer spectrometers. ESI-MS data were obtained by a Thermo quadrupole ion trap. Fluorescence imaging in zebrafish and cells was obtained by a using fluorescence microscope (MDG36, Leica and EVOS FL, Thermo Fisher Scientific). * Caution for the use of cyanide: Skin, respiratory, and eye protection is required. 

### 2.2. Synthesis of Sensor **DHADC** ((E)-3-((4-(diethylamino)-2-hydroxybenzylidene)amino)-2,3-dihydrothiophene-2-carboxamide)

3-Aminothiophene-2-carboxamide (1.1 mmol, 160 mg) and 4-diethylaminosalicylaldehyde (1 mmol, 200 mg) was dissolved in 12.0 mL of ethanol and blended for 8 h at 20 ℃. The deep orange powder was given by filtration and purified with ether. The yield: 72 % (230 mg); ^1^H NMR: 11.42 (s, 1H), 8.77 (s, 1H), 8.01 (s, 1H), 7.72 (d, *J* = 5.2 Hz, 1H), 7.62 (s, 1H), 7.50 (d, *J* = 8.8 Hz, 1H), 7.38 (d, *J* = 5.2 Hz, 1H), 6.36 (d, *J* = 8.8 Hz, 1H), 6.13 (s, 1H), 3.38 (m, 4H), 1.13 (t, *J* = 5.6 Hz, 6H); and ^13^C NMR: 163.74, 162.29, 159.94, 159.93, 152.67, 149.72, 132.52, 129.91, 120.31, 109.47, 105.12, 97.26, 44.50, and 12.99. ESI-MS for [**DHADC** (C_16_H_19_N_3_O_2_S) + H^+^]^+^: calcd, 318.13 (*m/z*); found, 318.21 (*m/z*). 

### 2.3. Fluorescence Titrations

1.0 mL of DMSO solvent was used to prepare a stock **DHADC** solution (1 × 10^−2^ mmol, 3.2 mg) and 3.0 μL of the stock solution was diluted to 2.997 mL bis-tris buffer (1 × 10^−2^ M, pH 7.0). 3–45 μL of zinc ion stock solution (0.1 M) dissolved in bis-tris buffer were added to **DHADC** (3.0 mL, 1 × 10^−2^ mM). After blending them for 20 sec, fluorescence spectra were obtained. Both **DHADC** and **DHADC**-Zn^2+^ showed no decomposition for 7 h in buffer condition.

For CN^−^, 1.0 mL of DMSO solvent was used to prepare a stock **DHADC** solution (1 × 10^−2^ mmol, 3.2 mg) and its eventual concentration (1 × 10^−2^ mM) was given by adding 3 μL of stock **DHADC** solution (1 × 10^−2^ M) into 2.996 mL MeCN. MeCN (1 mL) was employed to dissolve TEACN (tetraethylammonium cyanide, 33.1 mg (0.2 mmol)). 1.5–18 μL of cyanide (200 mM) were added to the compound **DHADC** (3.0 mL, 1 × 10^−2^ mM). After mixing them for 20 sec, fluorescence spectra were obtained. The titrations for Zn^2+^ and CN^−^ were conducted three times for the average.

### 2.4. UV-vis Titrations

3 μL of a stock **DHADC** solution (1 × 10^−2^ M) was transferred in fluorescent cell containing 2.997 mL bis-tris buffer. 6–42 μL of a zinc ion stock solution (0.1 M) in bis-tris buffer were added to the compound **DHADC** (3.0 mL, 1 × 10^−2^ mM). UV-vis spectra were obtained after blending them for 20 s.

For CN^−^, 3 μL of a stock **DHADC** solution (1 × 10^−2^ M) was added into 2.997 mL MeCN. 1.5–15 μL of this CN^−^ (0.2 M) in MeCN were added to the compound **DHADC** (3.0 mL, 1 × 10^–2^ mM). UV-vis spectra were obtained after blending them for 20 s.

### 2.5. Job Plots

350 μL of a stock **DHADC** solution (1 × 10^−2^ M) in DMSO was diluted to 49.65 mL bis-tris buffer for producing 0.07 mM. 0.3–2.7 mL of the diluted compound **DHADC** was added to fluorescent cells, respectively. 35 μL of a Zn^2+^ ion stock (1 × 10^−1^ M) solution in bis-tris buffer was diluted to 49.97 mL bis-tris buffer. 2.7–0.3 mL of the diluted zinc ion was added to each **DHADC** in fluorescent cells. The total volume of each fluorescent cell was 3.0 mL. Fluorescence spectra were obtained after blending them for 20 s.

For CN^−^, 0.6 mL of a stock **DHADC** (1 × 10^−2^ M) solution in DMSO was diluted to 29.4 mL MeCN/bis-tris buffer (95:5; v/v) to produce 2 × 10^−2^ M. 0.3–2.7 mL of the diluted compound **DHADC** was added to fluorescence cells, respectively. 30 μL of a CN^−^ stock solution (0.2 M) in MeCN was diluted to 29.97 mL MeCN/bis-tris buffer solution (95:5). 2.7–0.3 mL of the diluted cyanide was added to each **DHADC** in fluorescent cells. The total volume of each fluorescent cell was 3.0 mL. UV-visible spectra were obtained after blending them for 20 s.

### 2.6. Competition Tests

1 × 10^−2^ mmol of Al(NO_3_)_3_ or NaNO_3_ or Fe(ClO_4_)_2_ or In(NO_3_)_3_ or KNO_3_ or Ga(NO_3_)_3_ or M(NO_3_)_2_ (M = Ni, Pb, Ca, Mg, Cu, Mn, Co, and Cd), or Fe(NO_3_)_3_ or Cr(NO_3_)_3_ was separately dissolved in 1.0 mL of bis-tris buffer. 42 μL of each metal ion (1 × 10^−1^ M) was added into 3.0 mL bis-tris buffer to produce 140 equiv. Following the addition of 42.0 μL of a zinc ion stock solution (0.1 M), 3.0 μL of a stock **DHADC** solution (1 × 10^−2^ M) was added into the fluorescent cells containing each metal ion. Fluorescence spectra were obtained after blending them for 20 s.

For CN^−^, 0.2 mmol of NaNO_2_, tetraethylammonium salts of F^−^, I^−^ Cl^−^, and Br^−^, Na_2_S and tetrabutylammonium salts of N_3_^−^, H_2_PO_4_^−^, OAc^−^, SCN^−^, and BzO^−^ was separately dissolved in 1.0 mL of bis-tris buffer. 15 μL (2 × 10^−1^ M) of each anion was put into 3.0 mL MeCN/bis-tris buffer (95:5) to afford 100 equiv. Following the addition of 15 μL of TEACN solution (200 mM), 2 μL (1 × 10^−2^ M) of compound **DHADC** was added into the fluorescent cells containing each anion. Fluorescence spectra were obtained after blending them for 20 s.

### 2.7. ^1^H NMR Titrations

DMSO-*d*_6_ (2.8 mL) was used to dissolve compound **DHADC** (6.4 mg and 0.02 mmol) and 700 μL of **DHADC** was transferred to three NMR tubes, respectively. 0, 0.5, and 1 equiv of Zn^2+^ dissolved. in 1.2 mL of DMSO-*d*_6_ solvent were added to each compound **DHADC**. ^1^H NMR spectra were obtained after blending them for 20 s.

For CN^−^, DMSO-*d*_6_ (2.8 mL) was used to dissolve **DHADC** (6.4 mg, 0.02 mmol) and 700 μL of **DHADC** was transferred to four NMR tubes, respectively. 0, 0.5, 1, and 2 equiv of TEACN dissolved in 2.4 mL of DMSO-*d*_6_ solvent were added to each compound **DHADC**. ^1^H NMR spectra were obtained after blending them for 20 s.

### 2.8. Quantum Yields

The quantum yields of **DHADC**, **DHADC**-Zn^2+^ and **DHADC**-CN^−^ were determined with fluorescein (Φ_F_ = 0.92) in basic ethanol as a reference fluorophore. By using calibration curves of fluorescein and their absorption spectrum, the concentrations of fluorescein corresponding to each **DHADC**, **DHADC**-Zn^2+^, and **DHADC**-CN^−^ species were calculated and expressed as fluorescein-**DHADC**, fluorescein-**DHADC**-Zn^2+^, and fluorescein-**DHADC**-CN^−^. The quantum yields were calculated with the following equation [44].
$$\Phi_{F,S}={\;\Phi }_{F,R}\;\frac{A_{R}F_{S}}{A_{S}F_{R}}{\left(\frac{n_{S}}{n_{R}}\right)}^{2}$$
Φ_F_ is quantum yield, *A* is absorbance, *F* is the area of fluorescence emission curve, *n* is refractive index of the solvent, *S* is test sample, and *R* is a reference sample.

### 2.9. Quantification of Zn^2+^ in Real Samples

For fluorescent analysis in real samples, drinking water and tap water were obtained from our laboratory. The fluorescent analysis was carried out by adding 3.0 μL (10^−2^ M) of compound **DHADC** and 0.30 mL of a bis-tris buffer (10^−2^ M) to a 2.697 mL real sample solution having Zn^2+^. Solutions were thoroughly blended and remained at 20 °C for 5 min. Their fluorescence spectra were obtained.

### 2.10. Imaging in Live Cells and Zebrafish

In media containing 100 mg/mL streptomycin, the Eagle Medium, 10.0% fetal bovine serum, and 100 U/mL penicillin HeLa cells were kept. The cells grew in a humidified condition at 37.0 °C under 5% CO_2_. They were then put onto a 12 well plate (SPL Life Sciences, Pocheon, Gyeonggi-do, Republic of Korea) at a density of 1 × 10^4^ cells/0.1 mL, cells were seeded and then incubated at 37.0 °C for 20 h. For fluorescent imaging tests, cells were treated with compound **DHADC** (dissolved in DMSO, 3 × 10^−2^ mM) for 10 min, followed by the incubation of Zn(NO_3_)_2_ (dissolved in water, 5.0 mM) for 10 min. A EVOS FL fluorescent microscope was employed for imaging [emission 510 (±21) nm; excitation 470 (±11) nm].

The wild types of zebrafish (AB line) were incubated at 29 °C on a 14 h light/12 h dark cycle in E2 media (1 × 10^−3^ M MgSO_4_, 1 × 10^−3^ M CaCl_2_, 1.5 × 10^−4^ M KH_2_PO_4_, 1.5 × 10^−2^ M NaCl, 5 × 10^−5^ M Na_2_HPO_4_, 1 × 10^−4^ M KCl, 0.5 mg/L MB (methylene blue), and 0.7 mM NaHCO_3_ at pH 7.2). Six-day-old zebrafish were prepared for fluorescence bio-imaging in vivo. Zebrafish were fed with only 5 × 10^−3^ mM of **DHADC** in E2 media having 0.05% DMSO at 29 °C for 10 min. After the zebrafish were rinsed with E2 media to get rid of the remaining **DHADC**, the zebrafish were fed with the solution and had a wide range of concentrations of Zn^2+^ (20, 50, 100, and 200 μM) for 10 min at 29 °C. They were rinsed with E2 media again and then 0.01% ethyl-3-aminobenzoate methanesulfonate was added for the fixed orientation of zebrafish. A fluorescent microscope (MDG36, Leica) was employed to image all zebrafish (λ_ex_ = 450–490 nm. λ_em_ = 500–550 nm). By using Icy software, the mean fluorescence intensity was determined.

## 3. Results and Discussion

Probe **DHADC** was provided by the reaction of 4-diethylaminosalicylaldehyde and 3-aminothiophene-2-carboxamide in ethanol (72% yield, Scheme 1), and affirmed by ^13^C and ^1^H NMR and ESI-mass instrument.

### 3.1. Fluorescence Investigation of **DHADC** to Metal Cations

The sensing selectivity of **DHADC** was examined in the presence of various cations (Ni^2+^, Mn^2+^, Ga^3+^, Fe^3+^, Na^+^, In^2+^, Zn^2+^, Ca^2+^, Cd^2+^, Cu^2+^, Pb^2+^, Mg^2+^, Cr^3+^, Co^2+^, K^+^, Al^3+^, and Fe^2+^) in bis-tris buffer (Figure 1). By adding each metal ion (140 equiv), Zn^2+^ only exhibited a striking fluorescence increase (ca. 4500%) (λ_ex_ = 446 nm, λ_em_ = 508 nm). Instead, other cations did not increase the fluorescence. These outcomes illustrated that compound **DHADC** showed high discrimination towards Zn^2+^.

To examine the interaction between **DHADC** and Zn^2+^, fluorescent titration of compound **DHADC** to zinc ion was executed (Figure 2). The emission (508 nm) of compound **DHADC** steadily increased and indicated a maximum at 140 equiv. (λ_ex_ = 446 nm). Quantum yields (Φ) of 0.0003 (±0.0001) and 0.0135 (±0.0004) were determined for **DHADC** and **DHADC**-Zn^2+^ (Appendix A; λ_ex_ = 464 nm). Binding type of **DHADC** with zinc ion was also analyzed by UV-visible titration (Appendix A). By adding Zn^2+^ to compound **DHADC**, the peaks of 320 and 470 nm increased continuously, and that of 430 nm decreased. There were two definite isosbestic points (380 and 447 nm), meaning that the binding of **DHADC** to Zn^2+^ formed one product.

The binding process of **DHADC** and zinc ion was proposed to be a 1:1 interaction with the analysis of Job plot (Appendix A; λ_ex_ = 446 nm, λ_em_ = 508 nm) [45]. The 1:1 interaction of **DHADC**-Zn^2+^ was affirmed by the ESI-mass search (Appendix A). The mass data displayed that the peak of 458.00 (*m/z*) was reminiscent of [**DHADC**(-H^+^) + Zn^2+^ + DMSO]^+^ (calculated at 458.05). With fluorescent titration, the association constant (*K*) for **DHADC** with Zn^2+^ was given as 1.6 × 10^3^ (±31) M^−1^ by the equation of Benesi–Hildebrand (Appendix A) [46]. The constant was in the range of those (*K* = 1–10^12^) of previously announced probes for Zn^2+^ [47,48,49]. The binding process of compound **DHADC** with Zn^2+^ was further inspected by the titration of ^1^H NMR (Figure 3). With addition of Zn^2+^ (1 equiv.), the imine proton of 8.76 ppm was moved to downfield. At the same time, the protons of the thiophene moiety and the benzene ring were also moved to downfield. These results demonstrated that the N atom in the imine component and the O atom in the amide component may bind to Zn^2+^. No shift of the proton signals was monitored with the addition of more Zn^2+^ ions, which was indicative of a 1:1 binding of **DHADC**-Zn^2+^ species (Scheme 2). On the basis of the previous studies [34,50], the fluorescence turn-on mechanism of **DHADC** for Zn^2+^ might have the CHEF effect (chelation-enhanced fluorescence). During complexation of **DHADC** and Zn^2+^, the non-radiative transitions such as rotation and vibration were inhibited and the radiative transition was enhanced.

To test the practicable capability of compound **DHADC** as a Zn^2+^ detector, the competitive study was executed in a mixture of Zn^2+^ (140 equiv.) and various interfering ions (140 equiv.; Al^3+^, Pb^2+^, Ga^3+^, Fe^3+^, K^+^, In^2+^, Ni^2+^, Cd^2+^, Mg^2+^, Fe^2+^, Cr^3+^, Na^+^, Ca^2+^, Mn^2+^, Co^2+^, and Cu^2+^) (Appendix A; λ_ex_ = 446 nm, λ_em_ = 508 nm). Cu^2+^ fully interfered, and Fe^3+^, Cr^3+^, Fe^2+^, Co^2+^, and In^3+^ quenched 89%, 83%, 67%, 65%, and 22% of the fluorescence obtained with zinc ion alone. Therefore, the paramagnetic metal ions might be avoided for the applicability of the sensor in biological matrices. For practicable applications, the pH response of **DHADC**-Zn^2+^ was investigated at a wide variety of pH (2–12) (Appendix A; λ_ex_ = 446 nm, λ_em_ = 508 nm). **DHADC**-Zn^2+^ species exhibited a momentous fluorescence enhancement between pH 7 and 10. Therefore, Zn^2+^ could obviously be sensed by the fluorescent analysis with compound **DHADC** over the physiologically and environmentally important pH scope of 7.0–8.4 [51]. 

We established a calibration plot for the quantitative measurement of Zn^2+^ by compound **DHADC** (λ_ex_ = 446 nm, λ_em_ = 508 nm). Compound **DHADC** exhibited a satisfactory linearity between its intensity and the concentration of Zn^2+^, indicating that compound **DHADC** could be a possible choice for the quantitative measurement of Zn^2+^. With the use of 3 σ/slope [52], the detection limit was determined by 2.55 (±0.05) μM (Figure 4), which was much lower than the guideline (76 μM) recommended by the World Health Organization (WHO) [53,54]. To confirm the practicable ability of compound **DHADC** to Zn^2+^ in environmental samples, the samples of tap and drinking water were chosen (Table 1; λ_ex_ = 446 nm, λ_em_ = 508 nm). Acceptable recoveries and relative standard deviation (R.S.D.) values were obtained for the samples. Thus, compound **DHADC** can be operational for the measurement of Zn^2+^ in practical applications. 

In order to assess the sensing feasibility for biological applications of **DHADC**, we conducted fluorescent imaging experiments for sensing Zn^2+^ in living cells (Figure 5). We first incubated the HeLa cells with **DHADC** (30 μM) for 20 min. Then, the fluorescent emission in cells was not discovered without Zn^2+^. In contrast, the cells cultured with Zn^2+^ showed significantly increased fluorescence intensity. To further demonstrate the ability of **DHADC** in living organisms, the experiment for fluorescence imaging was carried out with zebrafish (Figure 6). When the zebrafish was incubated with **DHADC** (5 μM, a), there was no fluorescence signal. However, with increasing concentrations (20–200 μM, b–e) of Zn^2+^, the fluorescence signal gradually increased. By using Icy software, the mean fluorescent emission of the images was analyzed (Appendix A). The limit of detection was analyzed to be 21.44 (±2.6) μM. Thus, compound **DHADC** may be applied to intracellular sensing of Zn^2+^ in living organisms.

### 3.2. Fluorescence Studies of Compound **DHADC** to CN^−^

The fluorescence sensing capability of **DHADC** to a variety of anions in bis-tris buffer/acetonitrile solution (5:95) was examined (Figure 7; λ_ex_ = 459 nm, λ_em_ = 528 nm). The fluorescent spectra of **DHADC** with diverse types of anions (I^−^, S^2−^, H_2_PO_4_^−^, Cl^−^, N_3_^−^, F^−^, BzO^−^, Br^−^, OAc^−^, SCN^−^, and NO_2_^−^) showed very weak intensities. In contrast, there was a significant enhancement of fluorescence at 528 nm by adding 100 equiv. of CN^−^. These outcomes indicated that compound **DHADC** could have a potential function as a choosy fluorescence receptor for CN^−^.

To examine the influence of increasing levels of CN^−^ to **DHADC** solution, the fluorescence titration was carried out (Figure 8; λ_ex_ = 459 nm, λ_em_ = 528 nm). When the CN^−^ (0–120 equiv.) was added into **DHADC** solution, the fluorescence emission continuously increased at 528 nm and showed a maximum with 100 equiv. Quantum yields (Φ) of 0.0063 (±0.0004) and 0.1118 (±0.0003) were analyzed for **DHADC** and **DHADC**-CN^−^ (Appendix A). The binding character of compound **DHADC** with CN^−^ was inspected by UV-visible titration test (Appendix A). With the addition of cyanide to compound **DHADC**, the peaks at 315 and 475 nm increased consistently, and 390 nm decreased continuously with two definite isosbestic points (345 and 425 nm).

To investigate the binding mode of **DHADC** and CN^−^, Job plot analysis was performed (Appendix A; λ_ex_ = 459 nm, λ_em_ = 528 nm) [45]. This result showed a 1:1 complexation, which was affirmed by ESI-MS analysis (Appendix A). Addition of CN^−^ (1 equiv.) into compound **DHADC** exhibited the production of the [**DHADC** - H^+^]^−^ [*m/z:* 316.21; calculated at 316.11]. With the results of the fluorescent titration, the *K* value for **DHADC** with CN^−^ was given as 1.6 × 10^3^ (±50) M^−1^ (Appendix A). The detection limit (3σ/slope) was determined by 44.6 (±1.5) μM (Figure 9) [52]. 

To elucidate the detection process of compound **DHADC** with CN^−^, we carried out the titration experiments of ^1^H NMR (Appendix A). The proton of the hydroxyl component did not show up because of the possible inter or intra-molecular hydrogen bonds [55]. With the addition of CN (2 equiv.) to compound **DHADC**, all protons of the thiophene group and the benzene ring shifted to upfield. In contrast, one of the amide protons (H_3_) was shifted to downfield, suggesting that the H_3_ proton might hydrogen bond to CN^−^ or HCN species (Scheme 3). These outcomes implied that the negative charge generated from the deprotonation of compound **DHADC** by cyanide was delocalized through the **DHADC** [56]. No movement of the proton signals was detected with addition of more amounts of CN^−^ (>2 equiv.). On the basis of the previous studies and our experimental data [34,57,58], we can propose that the deprotonation of **DHADC** could cause the suppression of ICT (intramolecular charge transfer), which induces fluorescence turn-on of **DHADC**-H^+^ species. With the analysis results of ESI-mass, ^1^H NMR study and Job plot, the possible recognizing process of compound **DHADC** with CN^−^ was depicted in Scheme 3. 

To inspect the inhibition of different types of anions, the competitive tests were achieved and are shown in Figure 10 (λ_ex_ = 459 nm, λ_em_ = 528 nm). Compound **DHADC** was mixed with CN^−^ (100 equiv.) and a wide variety of anions (S^2−^, F^−^, BzO^−^, Cl^−^, SCN^−^, Br^−^, NO_2_^−^, OAc^−^, N_3_^−^, H_2_PO_4_^−^, and I^−^; 100 equiv.). Some inhibition was observed with F^−^, but its fluorescence was still discernible. These observations illustrated that compound **DHADC** may be an excellent selective fluorescence detector for CN^−^.

## 4. Conclusions

We demonstrated a unique fluorescent turn-on probe **DHADC** having a thiophene moiety. Compound **DHADC** could selectively sense Zn^2+^ and CN^−^ through fluorescence enhancement. Binding ratios of compound **DHADC** with Zn^2+^ and CN^−^ were proposed to be 1:1, with the analysis of ESI-mass data and Job plots. Detection limits for zinc ion and CN^−^ were 2.55 (±0.05) μM and 44.6 (±1.5) μM, respectively. The value for zinc ion was far below the standard (76 μM) of the WHO. Importantly, compound **DHADC** could be used to analyze zinc ion in water samples and to image zinc ion in both zebrafish and live cells. Additionally, compound **DHADC** could detect CN^−^ with little interference of competitive anions. Moreover, the detection processes of **DHADC** with Zn^2+^ and CN^−^ were proposed through ^1^H NMR titrations and ESI-Mass analyses. Therefore, the results observed in this study illustrate that **DHADC** can be a detector to selectively detect Zn^2+^ and CN^−^ by the fluorescent turn-on method in aqueous and living organisms.

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
