# Peer review of "A Novel Thiophene-Based Fluorescent Chemosensor for the Detection of Zn2+ and CN−: Imaging Applications in Live Cells and Zebrafish"

_sensors, 2019, doi:10.3390/s19245458_

Round 1
Reviewer 1 Report
In this study, Kim et al. present the synthesis of a new fluorescent molecule based on thiophene, and test is as a turn-on sensor for Zn+ and CN-. The work is presented well, the with the results seeming to back up their hypotheses.
My main question is whether the authors measured the extinction coefficient or photoluminescence quantum yield of their molecule? These are important characteristics of a new fluorophore, and it would be interesting to see how these values compare to comparable molecular fluorophores.
Minor comments:
Lines 184-186. It is stated that certain ions interfere with fluorescence signal. Could the authors add a comment about how this affects the applicability of the sensor in biological matrices?
Line 195. It is not obvious what the authors mean by “…under the standard (76 uM) guided by WHO…”. This should be explained better.
Line 213. The authors state “Limit of detection was analyzed to be 21.44 μM, which was still below WHO standard”. What is the meaning/relevance of this in the context of in vivo detection?
Author Response
Question 1: My main question is whether the authors measured the extinction coefficient or photoluminescence quantum yield of their molecule? These are important characteristics of a new fluorophore, and it would be interesting to see how these values compare to comparable molecular fluorophores.
Answer: We measured and mentioned the quantum yields of compounds DHADC, DHADC-Zn2+, and DHADC-CN- in lines 171 and 253. The results are shown in Fig. S1.
Question 2: Lines 184-186. It is stated that certain ions interfere with fluorescence signal. Could the authors add a comment about how this affects the applicability of the sensor in biological matrices?
Answer: We commented the following sentence in lines 201-202; the paramagnetic metal ions might be avoided for the applicability of the sensor in biological matrices.
Question 3: Line 195. It is not obvious what the authors mean by “…under the standard (76 uM) guided by WHO…”. This should be explained better.
Answer: As the reviewer 1 suggested, we revised it to “the guideline (76 μM) recommended by the World Health Organization” in line 211.
Question 4: Line 213. The authors state “Limit of detection was analyzed to be 21.44 μM, which was still below WHO standard”. What is the meaning/relevance of this in the context of in vivo detection?
Answer: Thank you for a good suggestion. We removed the expression.
Reviewer 2 Report
Kim et. al present a thiophene-Schiff base based fluorescent sensor to detect the presence of Zn ion and cyanide ion in live cell and zebrafish. Standard experiments to characterize fluorogenic probe for metal ions have been performed to support the conclusion. Strong and clear evidence was present in this manuscript though Zn ion sensors have been extensively explored and reported in the previous studies. Thus the major drawback of this work is its novelty. I support its publication in this journal after the author address the following concerns.
The author should provide comparison of previously reported Zn sensors and the uniqueness of their sensor. What is the advantage of the sensor present in this work? It has been reported that Schiff base is not stable under physiological conditions, in particular cellular environment. It will gradually decompose into aldehyde and causes cytotoxicity. The authors need to characterize the half life of the probe in buffer condition using HPLC or NMR. Is the probe stabilized by Zn ion binding? In scheme 3, the author proposed a mechanism to explain the why fluorescence changes upon binding to CN ions. However no experimental evidence support this conclusion. NMR experiment should be performed to support the deprotonation process upon ion binding.
Author Response
Question 1: The author should provide comparison of previously reported Zn sensors and the uniqueness of their sensor. What is the advantage of the sensor present in this work? It has been reported that Schiff base is not stable under physiological conditions, in particular cellular environment. It will gradually decompose into aldehyde and causes cytotoxicity.
Answer: We think that the advantage of our sensor can detect both Zn2+ and CN-. Therefore, we described the advantage in introduction. In addition, the sensor DHADC showed no decomposition for 7 h in buffer condition. We mentioned the result in line 76.
Question 2: The authors need to characterize the half life of the probe in buffer condition using HPLC or NMR. Is the probe stabilized by Zn ion binding?
Answer: Both DHADC and DHADC-Zn2+ showed no decomposition for 7 h in buffer condition. We mentioned the results in line 76.
Question 3: In scheme 3, the author proposed a mechanism to explain the why fluorescence changes upon binding to CN ions. However, no experimental evidence support this conclusion. NMR experiment should be performed to support the deprotonation process upon ion binding.
Answer: NMR experiment was performed and shown in Fig. S15.
Reviewer 3 Report
Kim et al. developed a thiophene-based fluorescent chemosensor for the detection of Zn2+ and CN–. They applied it to image Zn2+ in HeLa cells and zebrafish. The manuscript can be accepted after the following corrections.
Mention the full name of WHO in the abstract. Authors may update their literature survey for Zn2+ sensor e.g. Sensors 2019, 19(9), 2049, Spectrochimica Acta Part A 212 (2019) 222–231 and for CN– sensor: Sensors 2018, 18(7), 2219 and Dyes and Pigments 149 (2018) 764–773766. Provide all the graphs for quantum yield (QY) measurements of DHADC and DHADC-Zn2+ and DHADC-CN–. Also, write the details about the QY measurements including reference sample and equation. Inset figures in Figures 2 and 7 are not clear. Provide good image for Figure 3. How much Zn2+ was added for 1H NMR titration of DHADC? Mention it in Figure 3 caption. Write the Figure 4 caption properly mentioning the Figure numbers as a, b,.. Mention the excitation wavelengths for cell imaging study in Figures 4 and 5 captions. Provide a clear image for Figure 6. Mention the name of all other metal ions and anions in Figures 1 and 6 captions, respectively. Provide all the PL graphs for Figure S2. Write about the experimental procedure in detail for Job’s plot. The inset table in Figure S8 is not clear. The inset Figure in Figure S9 is not clear. What is the mechanism for fluorescence enhancement of DHADC upon addition of Zn2+and CN–? Explain the fluorescence mechanism elaborately with scheme and references. Why there are differences in concentrations of both DHADC and Zn2+ ions for fluorescent imaging of HeLa cells and Zebrafish? Mention the slit width for PL measurements in respective Figure captions. Authors mention that “When the zebrafish was incubated with DHADC (5 μM), there was no fluorescence signal (a)”. What does “(a)” indicate? There is no caution note for use of CN– salt. Check reference 41. Journal name is omitted.

Author Response
Question 1: Mention the full name of WHO in the abstract.
Answer: We revised it.
Question 2: Authors may update their literature survey for Zn2+ sensor e.g.
Sensors 2019, 19(9), 2049, Spectrochimica Acta Part A 212 (2019) 222–231
and for CN– sensor:
Sensors 2018, 18(7), 2219 and Dyes and Pigments 149 (2018) 764–773.
Answer: We cited the papers in refs. 9, 21, 37, and 44.
Question 3: Provide all the graphs for quantum yield (QY) measurements of DHADC and DHADC-Zn2+ and DHADC-CN–. Also, write the details about the QY measurements including reference sample and equation.
Answer: We added the details about the QY measurements in lines 127-133. All the graphs for quantum yield (QY) measurements of DHADC and DHADC-Zn2+ and DHADC-CN– are provided in Fig. S1.
Question 4: Inset figures in Figures 2 and 7 are not clear.
Answer: We made the Insets clearer.
Question 5: Provide good image for Figure 3. How much Zn2+ was added for 1H NMR titration of DHADC? Mention it in Figure 3 caption.
Answer: We provided a good image for Figure 3, and mentioned the amount of Zn2+ in Figure 3 caption.
Question 6: Write the Figure 4 caption properly mentioning the Figure numbers as a, b,.
Answer: We wrote the Figure 4 caption properly mentioning the Figure numbers as a, b.
Question 7: Mention the excitation wavelengths for cell imaging study in Figures 4 and 5 captions.
Answer: We added the excitation wavelengths in Figures 4 and 5 captions.
Question 8: Provide a clear image for Figure 6.
Answer: We provided a clear image for Figure 6.
Question 9: Mention the name of all other metal ions and anions in Figures 1 and 6 captions, respectively.
Answer: We mentioned the name of all other metal ions and anions in Figures 1 and 6 captions, respectively.
Question 10: Provide all the PL graphs for Figure S2. Write about the experimental procedure in detail for Job’s plot.
Answer: We described the experimental procedure in detail for Job’s plot in experimental section.
Question 11: The inset table in Figure S8 is not clear.
Answer: We provided the higher resolution image of the inset table in Figure S9.
Question 12: The inset Figure in Figure S9 is not clear.
Answer: We provided the higher resolution image of the inset table in Figure S10.
Question 13: What is the mechanism for fluorescence enhancement of DHADC upon addition of Zn2+ and CN–? Explain the fluorescence mechanism elaborately with scheme and references.
Answer: We tried to explain the mechanism for fluorescence enhancement of DHADC upon addition of Zn2+ and CN– by using the theoretical calculations. However, it was not successful. At present stage, it is difficult to explain the mechanism for the fluorescence enhancement.
Question 14: Why there are differences in concentrations of both DHADC and Zn2+ ions for fluorescent imaging of HeLa cells and Zebrafish?
Answer: The different concentrations of both DHADC and Zn2+ ions for fluorescent imaging of HeLa cells and Zebrafish were used to obtain the most obvious fluorescence turn-on.
Question 15: Mention the slit width for PL measurements in respective Figure captions.
Answer: We mentioned the slit width for PL measurements in respective Figure captions.
Question 16: Authors mention that “When the zebrafish was incubated with DHADC (5 μM), there was no fluorescence signal (a)”. What does “(a)” indicate?
Answer: We revised it.
Question 17: There is no caution note for use of CN‑ salt.
Answer: We mentioned caution note for use of CN- salt in Line 61.
Question 18: Check reference 41. Journal name is omitted.
Answer: We added the journal name in ref. 45.
Reviewer 4 Report
Major comments:
There is no mention of repetition of any experiments and no error bars included on almost all figures. Some figures are also missing units. I think the major challenge of this paper is the organization especially the methods section. Below are a few suggests for this. Suggest moving the scheme for DHADC synthesis into the intro and expanding on that in the intro. Line 69 repetitive wording could be eliminated by changing the description of DHADC to describe a stock DHADC solution that then can be referred to for the rest of the sections. “…DMSO solvent was used to make a stock DHADC solution (###). The stock DHADC solution was diluted by adding….” For methods sections Fluorescence titrations, UV-vis titrations, Job plots, and Competitive Tests make the Fe2+ and the CN- procedures separate paragraphs. Figure S8 is referred to in the methods section. Figures should be organized and numbered in the order they are mentioned in the article. Results “Fluorescence investigation of DHADC in Zn2+” section: I find it odd to read about the selectivity of a sensor before actually discussing the sensors response to one of the targeted analytes. Lines 63-67 would it be easier to include the spectra? You already have a supplemental section to put it in. Also could be referred to on line 144. I’m not sure it is appropriate to use “gained” when talking about obtaining data. Line 71 I think is the only time you refer to an actual buffer and the rest of the time you just say buffer. Make it clear if the bis-tris buffer is the only buffer used throughout the article. I think the results section could be expanded more. Specifically by including more numerical results like how much did the signal increase towards Zn+2 (line 151). Could use % signal enhancement calculations. Figure 3 I can’t read the x axis which would be useful when it refers to the certain peaks and their shifting. I would make sure the difference between the selectivity and competitive studies are clear. Selectivity is each ion tested individually while competitive is an ion with the targeted analyte at the same time.
Minor comments:
Line 15: change to “…has been…”
Line 26: change to “The design of chemosensors with high…”
Line 28: change to “…metal ions in the human…”
Line 29: should it be “second richest”?
Line 30: change to “…and in DNA-binding proteins.”
Line 31: change to “…in the body creates a wide variety…”
Line 32: change to “…chemosensors for selective sensing…”
Line 46: change to “…combined these two functional groups…”
Line 50: change to “…DHADC detected …”
Line 51: what is “respectively” referring to? It says both ions are turn on.
Line 57-58: reword
Line 71: change to “…of concentrated…”
Line 72: change to “…buffer were mixed with DHADC.” “put to” I don’t believe is the wording you want. Line 83 has the same.
Line 105: change to “…of a metal ion solution…”
Line 126-127: reword
Line 127: change to “…under 5%...”
Line 128: “bred” might not be the correct word to use.
Line 129-130: suggest rewording to “For fluorescent imaging tests, cells were treated with compound DHADC and Zn(NO3)2 and incubated for 20 mins.”
Line 132: “reared” may not be an appropriate word to use
Line 144: change to “72% yield”
Line 152: ex is a subscript with a weird space in front of it almost like something is missing. Same thing throughout
Line 153: I would take out the may be applied. “…DHADC shows high discrimination towards Zn2+”
Line 169: switch wording, “…association constant (K)…”
Line 170: constant can’t be “given”
Line 171: I can’t follow that sentence meaning.
Line 172: change to “…was further inspected…”
Line 183: large space between “Pb+2” and “Ga+3”
Line 194: odd wording “Detection limit was determined by…” as a suggestion for wording.
Table 1 title: “Determination of Zinc ion concentration”
Line 227: another reference to buffer without it being clear what it is.
Line 249: also not appropriate wording
Author Response
Question 1: There is no mention of repetition of any experiments and no error bars included on almost all figures. Some figures are also missing units. I think the major challenge of this paper is the organization especially the methods section.
Answer: We mentioned the repetition of fluorescence titrations in Line 82 and added the error bars in Figures 2, 7, S8 and S14. We also added some missing units.
Question 2: Suggest moving the scheme for DHADC synthesis into the intro and expanding on that in the intro.
Answer: We moved and mentioned Scheme 1 into the intro.
Question 3: Line 69 repetitive wording could be eliminated by changing the description of DHADC to describe a stock DHADC solution that then can be referred to for the rest of the sections. “…DMSO solvent was used to make a stock DHADC solution (###). The stock DHADC solution was diluted by adding….”
Answer: As the reviewer 4 suggested, we deleted the repetitive sentences.
Question 4: For methods sections Fluorescence titrations, UV-vis titrations, Job plots, and Competitive Tests make the Fe2+ and the CN- procedures separate paragraphs.
Answer: We separated the paragraphs of Zn2+ and CN-.
Question 5: Figure S8 is referred to in the methods section. Figures should be organized and numbered in the order they are mentioned in the article.
Answer: We corrected them.
Question 6: Results “Fluorescence investigation of DHADC in Zn2+” section: I find it odd to read about the selectivity of a sensor before actually discussing the sensors response to one of the targeted analytes.
Answer: We revised the expression to metal cations.
Question 7: Lines 63-67 would it be easier to include the spectra? You already have a supplemental section to put it in. Also could be referred to on line 144. I’m not sure it is appropriate to use “gained” when talking about obtaining data.
Answer: We revised them, and changed “gained” to “obtained”.
Question 8: I’m not sure it is appropriate to use “gained” when talking about obtaining data.
Answer: We revised “gained” to “obtained”.
Question 9: Line 71 I think is the only time you refer to an actual buffer and the rest of the time you just say buffer. Make it clear if the bis-tris buffer is the only buffer used throughout the article.
Answer: As the reviewer 4 suggested, we revised “buffer” to “bis-tris buffer”.
Question 10: I think the results section could be expanded more. Specifically by including more numerical results like how much did the signal increase towards Zn2+ (line 151). Could use % signal enhancement calculations.
Answer: As the reviewer 4 suggested, we used % signal enhancement calculations in line 163.
Question 11: Figure 3 I can’t read the x axis which would be useful when it refers to the certain peaks and their shifting.
Answer: We provided higher resolution image of Figure 3.
Question 12: I would make sure the difference between the selectivity and competitive studies are clear. Selectivity is each ion tested individually while competitive is an ion with the targeted analyte at the same time.
Answer: We think that we explained the difference properly.
Question 13:
Line 15: change to “…has been…”
Line 26: change to “The design of chemosensors with high…”
Line 28: change to “…metal ions in the human…”
Line 29: should it be “second richest”?
Line 30: change to “…and in DNA-binding proteins.”
Line 31: change to “…in the body creates a wide variety…”
Line 32: change to “…chemosensors for selective sensing…”
Line 46: change to “…combined these two functional groups…”
Line 50: change to “…DHADC detected …”
Answer: We revised all.
Question 14: Line 51: what is “respectively” referring to? It says both ions are turn on.
Answer: We revised it in line 52.
Question 15:
Line 57-58: reword
Line 71: change to “…of concentrated…”
Line 72: change to “…buffer were mixed with DHADC.” “put to” I don’t believe is the wording you want. Line 83 has the same.
Line 105: change to “…of a metal ion solution…”
Line 126-127: reword
Line 127: change to “…under 5%...”
Line 128: “bred” might not be the correct word to use.
Line 129-130: suggest rewording to “For fluorescent imaging tests, cells were treated with compound DHADC and Zn(NO3)2 and incubated for 20 mins.”
Line 132: “reared” may not be an appropriate word to use
Line 144: change to “72% yield”
Line 152: ex is a subscript with a weird space in front of it almost like something is missing. Same thing throughout
Line 153: I would take out the may be applied. “…DHADC shows high discrimination towards Zn2+”
Line 169: switch wording, “…association constant (K)…”
Line 170: constant can’t be “given”
Line 171: I can’t follow that sentence meaning.
Line 172: change to “…was further inspected…”
Line 183: large space between “Pb2+” and “Ga3+”
Line 194: odd wording “Detection limit was determined by…” as a suggestion for wording.
Table 1 title: “Determination of Zinc ion concentration”
Line 227: another reference to buffer without it being clear what it is.
Line 249: also not appropriate wording
Answer: We revised all more appropriately.
Round 2
Reviewer 1 Report
The author's have done a reasonable job of their modifications, and the article can be published.
Author Response
Question 1: Moderate English changes required.
Answer: We checked carefully the whole manuscript with a grammar-check program.
Reviewer 3 Report
Authors fail to provide PL graphs for Job’s plot. Authors should check the PLQY calculation. They also did not provide the absorbance spectra for PLQY measurements. DHADC+Zn2+ exhibited lower PL intensity than the DHADC in Figure S1(a). But in the manuscript, they mentioned that PLQY of DHADC+Zn2+ is higher than only DHADC. What is Fluorescein-DHADC and Fluorescein-DHADC+Zn2+ in Figure S1(a). Also, check the Fluorescein-DHADC+CN- in Figure S1(b). There are many reports for fluorescence enhancement upon addition of Zn2+ and CN–. Authors should provide logical explanation with references. Authors should check the reference 21.

Author Response
Question 1: Authors fail to provide PL graphs for Job’s plot.
Answer: We provided PL graphs for Job plot of DHADC with Zn2+ in Figure S3(a).
Question 2: Authors should check the PLQY calculation. They also did not provide the absorbance spectra for PLQY measurements. DHADC+Zn2+ exhibited lower PL intensity than the DHADC in Figure S1(a). But in the manuscript, they mentioned that PLQY of DHADC+Zn2+ is higher than only DHADC. What is Fluorescein-DHADC and Fluorescein-DHADC+Zn2+ in Figure S1(a). Also, check the Fluorescein-DHADC+CN- in Figure S1(b).
Answer: We provided the absorption spectra for PLQY measurements in Figure S1 and recalculated QY by referring to “Nature Protocols 2013, 8, 1535-1550”. We also added the detail on fluorescein in lines 129-132.
Question 3: There are many reports for fluorescence enhancement upon addition of Zn2+ and CN-. Authors should provide logical explanation with references.
Answer: We provided explanations for fluorescence enhancement upon addition of Zn2+ and CN- in lines 194-197 and 280-282, respectively.
Question 4: Authors should check the reference 21.
Answer: We corrected ref. 21.